# Provably safe Reinforcement Learning using Bender's Decomposition Oracles

## Abstract

One of the core challenges when applying reinforcement learning to solve real world problems is the violation of numerous safety, feasibility or physical constraints during training and deployment. We propose Bender's Oracle Optimization (BOO) that manages to achieve provable safety during both training and deployment, under the assumption that one has access to a representation of the feasible set, e.g., through a (possibly inaccurate) simulator or encoded rules. This method is particularly useful for cases where a simple (deterministic) model of the problem is available, but said model is too inaccurate or incomplete to solve the problem directly. We showcase our method by applying it to a challenging reward-maximizing stochastic job-shop scheduling problem, where we demonstrate a 17% improvement, and a nonlinear, nonconvex packing problem where we achieve close to globally optimal performance while improving the convergence speed by a factor of 800.

## 1 Introduction

Reinforcement Learning (RL) is a powerful technique for solving challenging Markov Decision Processes (MDPs) through interaction with an environment. This approach has recently shown impressive results in a wide variety of planning and control problems, such as scheduling (Bayliss et al., 2017), process planning (Floudas & Lin, 2005), robotics (oh Kang et al., 2023), and network design (Menon et al., 2013), as well as being extended towards more general problem sets, such as matrix factorization (Fawzi et al., 2022), quantum circuit optimization (Ruiz et al., 2024), or algorithm discovery (Mankowitz et al., 2023). One major obstacle for these approaches is that they need to guarantee that the solution to these planning problems is within a constrained subset of possible plans to qualify as a solution. Such constraints are often necessary for safety (e.g., a cargo ship should not be overloaded; the power grid should never produce a blackout, etc.) or feasibility reasons (e.g., a scheduler should not double-book appointments; a robot should not get stuck with an empty battery; a matrix factorization has to return the same matrix, etc.).

Classically, these constraints have been modeled using the framework of Constrained Markov Decision Processes (CMDP) (Altman, 1999), which introduce a cost function $c(x)$ whose expected cumulative (discounted) sum has to be below a threshold $C$:

$$\max_\pi \sum_{i=0}^{T} \gamma^i R(x_i) P(x_{i+1}|x_i, a_i) \pi(a_i|x_i) \tag{1a}$$

$$s.t. \sum_{i=0}^{T} \gamma^i c(x_i) \leq C \tag{1b}$$

While very popular, this modeling has some notable disadvantages, namely that hard constraints are difficult to model: Assume there is a set of states $X_C$ that should *never* be reached. Theoretically, we can model this in the above framework by placing $\forall x_c \in X_C : c(x_c) = \infty$, but in practice such an approach is not learnable if the illegal (or feasible) set is nontrivial. Feasible sets become nontrivial if the question $x \in X_C$ by itself is already computationally hard: For instance, consider the case of $X_C$ encoding an NP-complete problem such as a constraint satisfaction problems (CSP) or SAT. In those cases finding a single $x \in$ CSP is already NP-complete, which makes traditional black-box reinforcement learning highly intractable. These types of hard constraints appear frequently in the

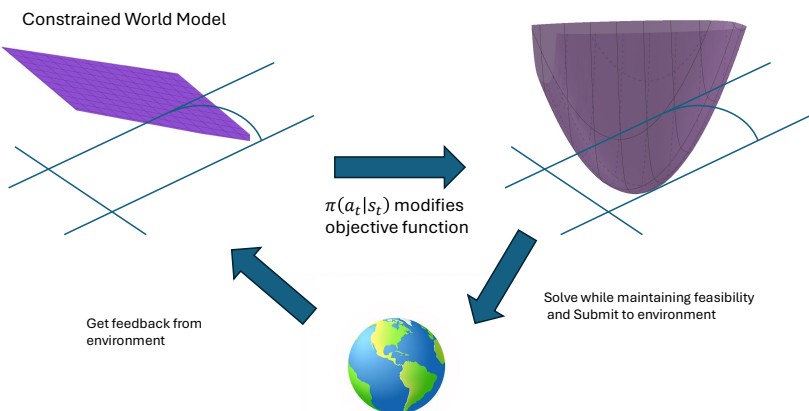

Figure 1: Overview of our method. We assume the existence of a (potentially NP-hard) world model over which we can optimize. Our policy $\pi$ modifies the objective function of the world model to give the desired behavior. The solution is given to the environment, which provides feedback (such as a stochastic events) to the world model, which can then be re-solved with the added information

real world: Consider the problem of autonomous driving where the car has to *always* stay on the road, or a program optimization tasks where the unoptimized and the optimized program have to describe the same input-output relationship. Another example for these are scheduling problems or routing problems where hard constraints (such as "no worker can do two jobs at once", "every item must be delivered in time", "avoid a blackout") have to be encoded to make sure that the result produced by policy are even valid plans to begin with.

This effect is even more pronounced when such *invariants have to be upheld during training* since the policy is not yet capable of controlling the state $x$ enough to stay within simple feasible set. In practice, upholding such guarantees during training promises to be very useful in the practical application of RL since it makes online-training of such models feasible. However, to guarantee absolute safety during training one has to make sure that the policy only affects the *quality* of a solution, not the *feasibility* of it. In current deep learning literature this is usually solved using ad-hoc reparametrizations (e.g., Fawzi et al. (2022); Bello et al. (2016)), which means such networks have to be manually designed to uphold invariants. Further, it is unclear whether such a reparametrization even exists for general NP-complete problems, such as CSPs[1].

In this work we showcase a general method that can solve arbitrarily constrained problems under the assumption that our policy has access to a constrained world model. By utilizing techniques from mathematical programming we guarantee the agent stays within the feasible region during inference *and* training, which allows for *fully online training without loosing safety guarantees*. We do this by proposing a universal parameterization that replaces the existing action set (e.g., "schedule job ABC in timeslot XYZ") with a new set of actions that correspond to cutting-planes inspired from the framework of the bender's decomposition. Instead of sequentially placing actions in a CMDP (Eq.1), this method sequentially modifies an optimization problem representing the set of "safe" plans (see Figure 1) to find high value solutions. This way, we can generate safe trajectories over the CMDP during training and inference, at the cost of needing an approximate world model. We also remain scalable by moving the complexity of strict feasibility preservation into a dedicated solver such as SCIP (Bestuzheva et al., 2021) without sacrificing the expressivity of our neural network. We showcase our method by applying it to a challenging reward-maximizing stochastic job scheduling problem, and a challenging nonconvex nonlinear discrete packing problem.

## 2  RELATED WORK

From the point of view of constrained reinforcement learning, prior work mostly considers the setting of feasibility budgets where the constraint violations have to stay below a certain threshold

---

[1]If $P \neq NP$ then such a parametrization cannot exist for certain problems.

(see Eq. 1). Constrained Policy Optimization (CPO) (Achiam et al., 2017) extends the popular PPO (Schulman et al., 2017) algorithm and supports constraints by descending inside the intersection of a trust region and the feasible set, using recovery steps when the policy is outside the feasible set. This method does not have a guarantee to be safe during training and cannot natively handle hard constraints. Chow et al. (2015) uses a primal-dual approach where the primal (policy) parameters are learned jointly with the Lagrangian-dual multipliers. They also consider cumulative costs, rather than hard constraints, and use the *conditional value at risk (CVaR)* framework to keep the learned policy within a set of low-risk policies. Another sometimes competitive approach is penalizing constraint violations with large negative values inside the reward, such as Fixed Penalty Optimization (FPO) (Achiam et al., 2017). Tessler et al. (2018) uses a more sophisticated version of FPO by dynamically adjusting the penalty parameter $\lambda$ during optimization. However, neither of these models handle hard-constraints or even training-time constraints.

Perhaps the closest work to ours is Dalal et al. (2018). They consider exploration inside a continuous space where safety is guaranteed by re-projecting any action into a feasible set of safe actions. While they consider hard constraints, they can only operate in continuous action spaces. Continuous action spaces are often significantly easier to solve from a safety point of view as one can smoothly route around critical areas. This is in stark contrast to combinatorial problems, where one may need to plan many steps ahead to be able to plan around dangerous actions. Similar to our method, they also delegate their safety constraint to a classical solver (in their case a Quadratic Programming (QP) solver) to compute the projection onto the feasible set. Our method has the advantage of not being limited to QP-solvers and being able to deal with combinatorial settings.

From the point of view of combinatorial optimization and RL, one seminal work to mention is from Bello et al. (2016), who consider both a solver for the travelling salesman problem (TSP) and Knapsack problem that shows strong performance on solving both these classical problems. However, they do not consider learning from a stochastic or nonlinear environments. Their method also needs specialized parametrizations (i.e., so-called pointer networks (Vinyals et al., 2015)) for every problem type which does not even exist for many problem types. For instance, Bello et al. (2016) uses the fact that the TSP instances they consider live on a fully connected graph, meaning that one can arbitrarily pick any order of nodes and will still get a possible (but perhaps very bad) tour. If one had a sparsely connected graph, Bello et al. (2016)'s method would no longer work as picking certain node orders can get the agent into a dead-end[2].

## 3   BACKGROUND: BENDER'S DECOMPOSITION

We frame our solution around a classical optimization concept known as the "(generalized) Bender's decomposition" (Geoffrion, 1972). Consider the following optimization problem

$$\min_{x,y} f(x, y) \tag{2}$$

$$s.t. \ g(x, y) \leq 0 \tag{3}$$

$$x \in X, y \in Y \tag{4}$$

where we assume $y$ to be vector of *complicating* variables. A complicating variable is a variable that, if fixed, makes the rest of the optimization problem much easier. For instance the problem $\min_{x,y}(\sin(y) - x)^2$ becomes trivial if we first fix $y$ to any value.

Bender's decomposition splits this optimization problem into a *master problem* and a *subproblem*. The master problem proposes solutions to the problem in $y$, ignoring the impact of the choice of $x$. The subproblem then uses the solution $y$ from the master to solve for the remaining variables $x$. Based on the value and feasibility of the subproblem we then add additional constraints into the master problem and repeat the optimization with the additional constraint.

Specifically, we can distinguish *feasibility constraints*, which remove items from the master problem that do not lead to a solvable subproblem, and *optimizing constraints* which manipulate the objective function of the master problem to steer it towards better solutions. To control the objective function, one classically adds an auxilliary variable $\varphi$ that is lower bounded by cutting information from the

---

[2]In fact, it might be the case that such a tour does not exist which is an undetectable case for Bello et al. (2016). Generally, deciding whether such a tour exists is already NP-complete (Held & Karp, 1965)

subproblem. Schematically, the master problem looks like

$$\min_y f(y) + \varphi \tag{5}$$

$$s.t.\ g(y) \leq 0 \tag{6}$$

$$\varphi \in \mathcal{O}(x, y), y \in \mathcal{F}(x), x \in X, y \in Y, \tag{7}$$

where $f(y)$ and $g(y)$ are lower bounds in $x$ to $g(x, y)$ and $f(x, y)$ respectively, and $\mathcal{O}, \mathcal{F}$ are additional constraints that are generated by solving a subproblem (see Geoffrion (1972)). For the sake of this work, we will only consider optimality constrains $\mathcal{O}(x, y)$. One can show that for many problems this process will yield the same result as the original problem[3], but due to the decomposition this model can usually be solved significantly faster.

## 4 BENDER'S ORACLE OPTIMIZATION

Ordinary optimality cuts have the form of

$$\varphi \geq z(x^*) + \lambda^T \nabla_x g(x^*, y^*)(x - x^*), \tag{8}$$

where $z(\cdot)$ is the result of the subproblem $z(x^*) = \min\{d^T y : g(x, y) \leq 0, y \geq 0\}$ conditioned on the solution $x^*$ of the master problem, $\lambda$ is the optimal dual solution, $y^*$ is the solution to the subproblem, and $\varphi$ is a helper variable that is added to the objective $\max c^T x + \varphi$. This has a nice interpretation of placing a lower bound based on the main problem on the linearization of the subproblem (see, e.g. Geoffrion (1972)).

Instead of solving a subproblem, we propose a "Benders Decomposition Oracle" that directly learns a scalar corresponding to the bias $b = z(x^*) + \lambda^T \nabla_x g(x^*, y*)x^* \in \mathbb{R}$, and a vector corresponding to the linear weight $w = \lambda^T \nabla_x g(x^*, y*) \in \mathbb{R}^d$ without explicitly constructing and solving the underlying optimization problem(s). Both of these values are learned end-to-end using reinforcement learning from feedback by a simulator. This means that instead of creating a CMDP across the "time" dimension where actions are iteratively unrolled, we create an MDP across a "cutting plane" dimension which iteratively places more constraints on the model until a high-valued plan is found. Specifically, we train a policy $\pi(b, w|s)$ such that after applying a number of $K$ cuts to the program, the resulting solution $x^*$ performs better according to some stochastic, and possibly nonlinear objective. We call $\pi(b, w|s)$ the "Bender's Decomposition Oracle" (BDO) and the algorithm resulting in the use of BDOs "Bender's Oracle Optimization" (BOO).

Specifically our MDP $(\mathcal{S}, \mathcal{A}, \mathcal{T}, \mathcal{R})$ has the state space $\mathcal{S}$ consisting of (a relevant subset of) the original state and action space, as well as additional external features (see below), the action space $\mathbb{R} \times \mathbb{R}^{|x^*|}$ consisting of the vector $w$ and the bias $b$, and the transition function given by some existing solver like SCIP (Bestuzheva et al., 2023) or IPOPT (Wächter & Biegler, 2005) combined with possibly a simulator injects stochastic information, and the reward function $\mathcal{R}$ given by some real-world metric. Notice that the resulting MDP does *not* need constraints since they are automatically parameterized into the solver. This parameterization is especially appealing if the action and state-space coincide, such as in automatic planning or constraint satisfaction. Our agent $\pi(b, w|s)$ predicts the coefficients of a new cut $c_i$:

$$k_i \quad : \quad \varphi \geq b + w^T x \tag{9}$$

which is added to the optimization problem. The augmented model

$$\max\{c^T x + \gamma \mid Ax \leq b, k_0, k_1, \dots\}, \tag{10}$$

is solved and the solution is passed back to $\pi$ to add another cut.

We parameterize $\pi$ as a Graph Neural Network (Kipf & Welling, 2016) connecting *variable nodes* with *constraint nodes*. Every variable node contains its upper and lower bound, the value of the variable in the current solution, as well as relevant additional information based on the task (for instance a "job" variable for a scheduling problem may contain the type of job). The constraint nodes contain the constraint bias and value of the constraint. For simplicity, we only consider linear constraints in this work, but this is not a limitation of the technique, as we could replace our MILP

---

[3]The exact conditions under which the bender's decomposition will give the same results as the original problem can be quite technical and are not too relevant for our usecase (for details, see e.g. Geoffrion (1972))

solver with a more general MINLP solver. The graph is built by connecting every variable to every constraint that contains that variable. The edges between variables and constraints are weighted by the coefficient of the variable in the constraint.

Modeling safe reinforcement learning this way has a couple of key advantages over both ordinary reinforcement learning and ordinary stochastic optimization. Regarding the former, notice that we can trivially uphold arbitrary safety and feasibility guarantees on the solution $x^*$ by explicitly constraining them in the master problem. This is particularly important in cases where guaranteeing feasibility is highly nontrivial, or when learning *during deployment*. Further notice the scalability advantages of this method: Since we can delegate feasibility constraints to highly advanced optimizers, we can quickly solve highly constraint problems.

From the point of view of stochastic programming, we have the advantage that our method can incorporate arbitrary (nonconvex) nonlinear and stochastic effects inside its cut-generating function: Notice that the policies input state $s \in \mathcal{S}$ can include both information from the master solution $x^*$, but also *external* information, which helps us to learn from the environment as in every other RL problem. For instance, in the case of job scheduling, one can include the type of job, likelihood of a person getting sick, expected time taken for the job, expected profit for the job, etc. into the estimation of the cut. The impact of these features is generally hard to model classically or it introduces a high degree of nonlinearity into the optimization. The RL framework we propose only adds affine-linear constraints to the problem (Eq. 8), meaning that every linear program stays linear, every convex problem stays convex, etc. This makes guaranteeing feasibility very fast since we can take advantage of high quality specialized solvers for e.g. mixed-integer linear programming.

## 5 EXPERIMENTS

We benchmark our solver on two problems. First, we study learning a nonlinear and nonconvex objective function over a nontrivial feasible set, but without considering stochasticity. Since we can set this problem, we can compare against the global optimum as found by the SCIP global nonlinear optimizer (Bestuzheva et al., 2023). We also utilize this problem to showcase an interesting, while not unexpected property of our method: Our solver is capable of learning a parametrization of the problem that is *significantly* faster to solve than the true parametrization.

Second, we consider a stochastic job scheduling problem, where the objective is to maximize the profit of a set of jobs, each consisting of a set of operations that have to be completed in order, within a limited time. A job only receives profit if all its operations are completed in the correct order by the time of completion. We add stochasticity to the problem, by having a set of task-types that determine how likely a job is delayed (forcing replanning) and how much profit is to be made by completing the job. This means our agent has to learn a complex risk-reward tradeoff, while also having to produce feasible job-scheduling plans.

### 5.1 NONCONVEX CONSTRAINED PROBLEM

To estimate the ability of our method to recover a nonlinear objective function over a constrained set, we consider the following problem

$$\max_x x^T A x + b^T x + c \tag{11a}$$

$$k^T x \leq p \tag{11b}$$

$$x \in \{0, 1\} \tag{11c}$$

where $A$ is a random positive semidefinite matrix, $b$ and $k$ are random vectors, and $k$ is a random constant and $c$ is an offset always set to $c = 1$. This type of problem is frequently found in economics where many problems can be reduced to convex maximization over binary variables subject to linear constraints (see Zwart (1974)). There are also applications to machine learning like, for instance, non-negative sparse PCAs (Zass & Shashua, 2006) or feature selection (Mangasarian, 1996).

We use this model as input to the global optimizer in SCIP (Bestuzheva et al., 2023), but *hide* the objective for our RL agent. The goal of our agent is to find optimizing cuts, such that the found $x$ maximizes the hidden $x^T A x + b^T x + c$ while staying feasible. Notice that this problem is *not convex* since we maximize over a convex function rather than minimize (see, e.g, Zwart (1974)).

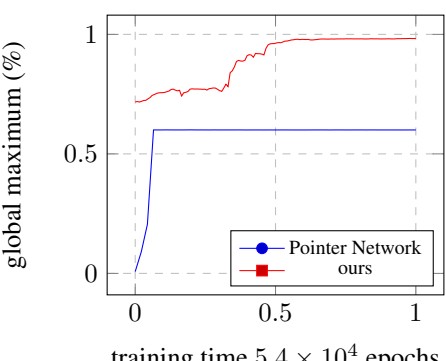 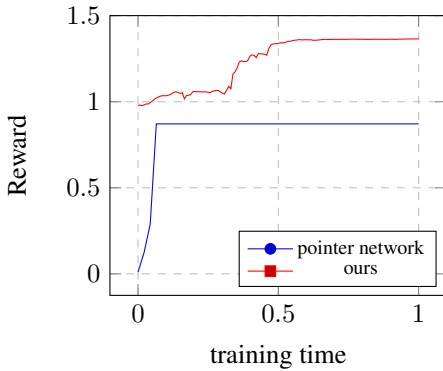

Figure 2: Validation performance of the pointer network baseline compared to ours. The x-axis is normalized towards the number of problems since both methods use different numbers of steps per problem (our method uses dramatically fewer steps).

The reason we choose to use a positive semidefinite (psd) matrix is because it allows us to give the following two features: First, we give the diagonal value of $a_{i,i} \in A$ for every variable $x$. Second we give the row/column sum $\sum_{i=0}^{N} a_{i,j}$ for every variable. This should allow the method to estimate, for instance, the eccentricity of the corresponding metric. In addition to those two features, we also supply the current solution $x_k$ and the instance parameters $b, k, p$. Generally, this is *insufficient to reconstruct the entire objective* function. This means the problem is a constrained Partially Observable Markov Decision Process (CPOMDP), where the model has to gather additional information from the found solutions $x_k$.

As a baseline, we utilize a variant of the pointer network (Vinyals et al., 2015) used in Bello et al. (2016) with the difference that instead of a simple unstructured RNN (Schmidt, 2019), we use exactly the same GNN backbone as in our method to make sure no method is disadvantaged by a smaller/larger network or different data availability. To accomplish this, we use a softmax over all GNN nodes that correspond to variables with the already chosen variables being masked out. Selection stops when either the constraint $k^T x \leq p$ would be exceeded by the chosen action, or when the model chooses to use a dedicated "stop selection" action.

We further compare against a naive baseline where we optimize Eq. 11a by linearizing the objective around 0. This is comparable to what one would obtain if one directly tried to optimize the MIP without knowing that it had nonlinear correlations in its objective. Notice that both this linearized model and our BOO model can be efficiently optimized with LP-solvers, while the original objective has to use much more complex MINLP solvers. As our reward we compare the quality of the solution found by our policy against its linearization:

$$R = \frac{x_\pi^T A x_\pi + b^T x_\pi + c}{x_b^T A x_b + b^T x_b + c},$$

(12)

where $x_\pi$ is the solution found by BOO, and $x_b$ is the result found by maximizing the linearized objective. $R > 1$ means our model exceeds the naive baseline, while $R < 1$ implies the model is worse than the linearized objective. This reward is used both for our method and the pointer network baseline. We report both the reward and the percentage towards the global optimum by both methods.

As we can see in Fig. 2, our method manages to reach almost the $100\%$ of true objectives value after roughly half the exploration budget has been reached. The pointer network quickly reaches a saturation level of roughly $60\%$ of the global maximum. It is worth noting that the x-axis of Fig. 2 is normalized based on the number of trials. This is necessary since pointer networks need 1 step per placed item, while our method scales with the number of cuts $K$. To make sure both methods have the same effective training budget, we fix the number of environment deployments, rather than the number of steps (our method only needs 1% of the steps the pointer networks need).

Looking into Table 1, we can also see that our method tends to find solutions orders of magnitude faster than the MINLP solver that knows the objective function with minimal loss in quality. This is

not entirely unsurprising as Bender's decomposition is fundamentally a way of speeding up MINLP problems (see Section 3 or Geoffrion (1972)), but it is nevertheless interesting to see that this property translates to black-box learning of objective functions. We also noticed that as our method improves, it tends to learn policies that find optima faster (see Appendix A). This opens up an interesting secondary usecase where such a policy is trained directly with the goal of quickly finding high quality MINLP solutions.

One advantage of a properly constrained RL agent is that one can train during deployment without having catastrophic failures in safety. Therefore we also report the regret (i.e., the area under the performance curve in Fig. 2) one would expect when training this agent online in Table 1. As one can see our agent outperforms the pointer network by close to $4\times$.

## 5.2 Scheduling Problem

We use a model loosely based on the time-indexed scheduling problem (see e.g., Ku & Beck (2016)). Specifically, we consider the problem of finding a schedule that maximizes returns within a fixed timeframe. In our setup, we consider 3 different machine types, where each machine has $M$ duplicates. We sample $J$ jobs that have a randomly sampled expected completion time for each machine. The machines have to be worked on in order: First machine 1, second machine 2, third machine 3. A job only pays out its profit, if all of its operations where completed in time and in the right order.

This gives us the following set of constraints on our policy: Let $y_j$ be a binary indicator of whether job $j = 1, \ldots, J$ is worked to completion, $x_{m,j,t}$ be the binary indicator of whether job $j$ is scheduled on machine $m = 1, \ldots, M$ at timestep $t = 1, \ldots, T$. The set of feasible schedules is given by:

$$\sum_{t=1}^{T} x_{m,j,t} \leq 1 \qquad \forall m = 1 \ldots M \forall j = 1 \ldots J \tag{13a}$$

$$y_j \leq \sum \sum_{t=1}^{T} x_{m,j,t} \qquad \forall m = 1 \ldots M \forall j = 1 \ldots J \tag{13b}$$

$$\sum_{t=0}^{T} (t + \text{jobtime}(j)) x_{m,j,t} \leq T \qquad \forall m = 1 \ldots M \forall j = 1 \ldots J \tag{13c}$$

$$\sum_{j=1}^{J} \sum_{t'=t-\text{jobtime}(j)+1}^{t+1} x_{m,j,t'} \leq M \qquad \forall m = 1 \ldots M \forall t = 1 \ldots T \tag{13d}$$

$$\sum_{t=0}^{T} (t + o(j, m-1)) x_{m-1,j,t} \leq \sum_{t=0}^{T} t x_{m,j,t} \qquad \forall m = 2 \ldots M \forall j = 1 \ldots J \tag{13e}$$

$$\sum_{t=0}^{T} x_{m-1,j,t} \geq \sum_{t=0}^{T} x_{m,j,t} \qquad \forall m = 2 \ldots M \forall j = 1 \ldots J \tag{13f}$$

$$\text{jobtime}(j) = \sum_{m=1}^{M} o(j, m) \tag{13g}$$

$$y_j, x_{m,j,t} \in \{0, 1\} \tag{13h}$$

Table 1: Comparisons of pointer network and our method over our validation set. We showcase the quality of the found solution as a percentage of the globally optimal value, and the time needed to find that solution (the global MINLP time is for reference).

|  | % global maximum | time policy | expected regret |
| --- | --- | --- | --- |
| ours | **0.98** | **0.07**s | **0.11** |
| pointer network | 0.60 | 1.10s | 0.43 |
| global MINLP | 1.0 | 60.19s | 0.0 |

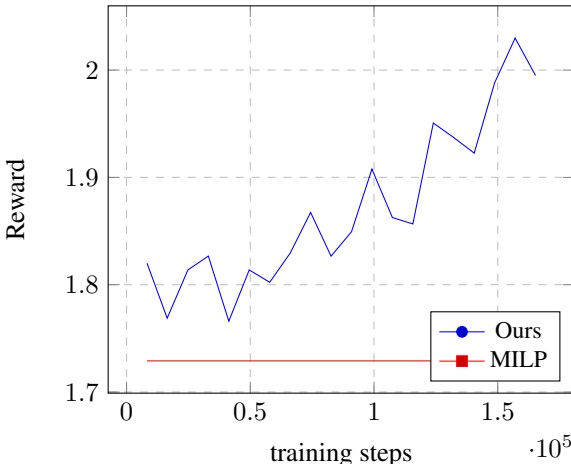

Figure 3: Performance of our model compared to optimal solutions provided by a MILP solver. The performance is shown over a *unseen* validation set

where $T$ is the global timelimit, and $o(j, m)$ is the time job $j$ takes on machine $m$. Equation (13a) makes sure every operation is only scheduled once, eq. (13b) sets the auxiliary variable $y_j$ denoting whether a job $j$ is completed in time, eq. (13c) makes sure that all scheduled operations complete within the timelimit, eq. (13d) prevents two operations being scheduled on the same machine simultaneously, eq. (13e) makes sure that operation $m$ of job $j$ happens after operation $m - 1$, and eq. (13f) makes sure that if machine $m$ is scheduled, machine $m - 1$ also has to be scheduled. This is a highly constrained MILP problem, meaning that randomly generating a plan $x_{m,j,t}$ is almost always going to be infeasible according to the constraitns eq. (13).

Within this feasible set, every assignment of $x_{m,j,t}$ corresponds to a plan that is expected to be feasible. After our solver decides on a plan, we apply that plan to our environment by simulating from $t = 0$ to $t = 12$ months. During that time, we randomly extend the time taken for a scheduled operation $o(j, m)$ by between 1 and 3 months. The likelihood a job is delayed depends on the job class $C(j)$, which is randomly sampled and given to our policy as a feature.

After a job is delayed (and therefore the existing plan is violated), we re-schedule with the newly added constraint. The profits are similarly hidden, but also depend on the job class $C(j)$, such that a riskier job obtains a higher payoff. This gives a highly complex risk-reward tradeoff where one has to balance risky but high profit jobs against lower risk, but lower profit jobs. For our experiments we choose $T = 12$, $J = 200$ and $M = 4$.

As a reference value, we solve this model as a baseline to $\max \sum_{j=1}^{J} y_j$, which can be seen as an uninformative prior, where all stochastic and (nonlinear) profit functions are ignored, in favor of simply packing the schedule as tightly as possible. We do not use the true job-rewards as the objective function since that would cause the MILP to plan all high reward, but also high-risk jobs (which is highly suboptimal). In practice, a tight scheduling MILP like the one we use performs significantly better than a job-reward maximizing MILP. We run both the baseline and our method and evaluate them using the environment, re-planning when necessary. This corresponds to a classical solution where the problem is modeled as a deterministic mixed-integer program.

Since, to our knowledge, no solver for this challenging stochastic planning problem exists, we compare ourselves against a classical MIP formulation that plans optimally with the information it has, and re-plans in the case of a stochastic event. This means the baseline plan is optimal up-to the unknown information introduced by the stochasticity and unknown profit per completed job. We also tried to apply the pointer network method to solve this problem, but the policy was unable to learn anything that performs better than a random policy. This is mostly because the model is unable to maintain temporal constraints on operation $i + 1$ having to be scheduled after operation $i$.

Instead of comparing against the pointer network, we compare against the solution found by the optimal MILP solution *without* considering stochastic effects and knowledge of the true value of

Table 2: Comparison of our method against an optimal MILP solver (higher is better).

|  | Reward@$0.5 \times 10^5$ | Reward@$1.0 \times 10^5$ | Reward@$1.5 \times 10^5$ |
| --- | --- | --- | --- |
| Ours | **1.80** | **1.87** | **2.03** |
| MILP | 1.73 | 1.73 | 1.73 |

each job. This means we compare our learning based method on a stochastic environment against an optimal agent inside a deterministic environment. For our agent to beat the baseline, it has to both be able to deal with stochastic effects, and has to learn the true value of completing a job. For this, we set up the job values as the likelihood of a job being interrupted, i.e., if a job has probability 0.9 of being delayed at any specific point in time, the reward for completing it is 0.9. This gives a natural risk-reward structure, where riskier jobs yield more reward.

The results for this can be found in Fig. 3 and Table 2. As we can see our method quickly exceeds the performance of the greedy MILP solver. Since our method always returns a valid schedule, this method can be used as a drop-in replacement for traditional MILP solvers when feedback from the environment is available. Since our method can be trained during deployment, it makes sense to also consider the advantage of our method against the baseline. Our method offers an expected improvement over the training interval (Fig. 3) of $\frac{\int \text{ours}(t)dt}{\int \text{base}(t)dt} \approx 8.2\%$. Note that this metric depends heavily on the training time since longer training times mean the model spends more time in the RL-optimized region.

## 6  LIMITATIONS

The main limitation of this method is the need for a representation of the constraints and decision variables. In general reinforcement learning these types of model may be hard to get, but we would argue that in cases where hard constraints are demanded during training one generally has access to such a model. This is because if one wants to have any hope of being absolutely safe during training, one needs to have a notion of safety *before* a single step is taken. Therefore we would argue that having access to a constrained model is not fully unrealistic in safety critical or high complexity scenarios. One can also learn a model of the constraint set (like Eq. 13) from data, but as this is a completely orthogonal problem from acting inside such a model, we do not discuss this here.

Learning a safe model can also be viewed as learning feasibility cuts (see Section 3), which we do not explicitly cover in this work. However, extending our framework to this would be relatively straight forward, as one can simply train a second policy $\pi^{\text{feas}}$ that predicts feasibility cuts, rather than optimality cuts. The reason we do not cover this here is because it is unclear how one should assign credit for those cuts. This is especially an issue for stochastic environments where an instance might be feasible in the realization of the random variables that was actually sampled, but might be infeasible for almost all other instances. In those cases one has to answer the question of whether such an instance should be judged as feasible or infeasible. For this reason, we leave the issue of credit assignment for feasibility cuts for future work.

## 7  CONCLUSION

We propose a generalized method for enforcing arbitrary (known) constraints in highly constrained reinforcement learning problems. Our method is able to enforce arbitrary hard constraints during both training and inference, allowing for more flexible utilization of our reinforcement learning in (safety)-constrained environments. Due to the utilization of affine-linear cuts, we can use highly efficient solvers which allows us to scale to complex combinatorial problems which are usually out of reach for reinforcement learning.

We showcase the abilities of our method in a synthetic combinatorial environment, and a job-scheduling problem. Our method shows superior performance over both a MILP and neural-network baseline, while offering drastically faster convergence compared to a MINLP solver, in cases where an analytical expression exists. To our knowledge this is the first reinforcement learning method that allows arbitrary constraints to be enforced during training and inference.

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

## A   Optimization Speed over time

We find that BOO implicitly learns to find solutions more efficiently. We assume this is because we impose a 60s time budget on finding solutions during training time, since this is the expected solving time for our problem class. This might implicitly regularize found policies towards simpler solutions as more complex solutions run the risk of not being solvable to global optimality within

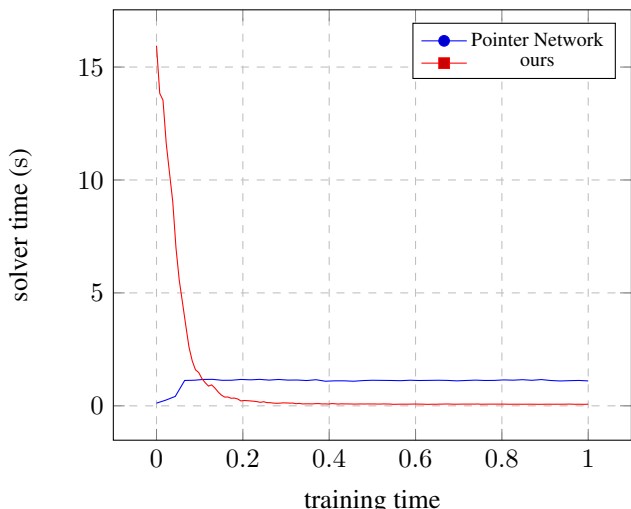

Figure 4: Time taken to find a solution.

the time budget. We assume this effect could be increased by explicitly including training time in the objective, but investigating this is left for further research.

The reason that the pointer network increases in time to find a solution is because the model learns to take advantage of the existing budget given by $k^T x \leq b$, which means it can place more items $x_i = 1$ into the feasible set, which then implies that one has to roll out the RNN over more timesteps, leading to slower inference.

