# OpenReview forum: "Provably safe Reinforcement Learning using Bender's Decomposition Oracles"
_ICLR.cc/2025/Conference — ICLR 2025 Conference Withdrawn Submission_

### Official Review · Reviewer_SK4U · 2024-11-03

**Soundness:** 2
**Presentation:** 3
**Contribution:** 2
**Rating:** 3
**Confidence:** 4

**Summary:**

This paper presents a learning-to-optimize approach based on RL and Bender's decomposition.

**Strengths:**

It combines learning to learn by RL and Bender's decomposition. Specifically, the action from the RL policy is the weight/ bias to generate new cuts in Bender's decomposition.

**Weaknesses:**

-While the combination is interesting, all the modules seem to come from standard off-the-shelf studies.

-The experiments are not sufficient to demonstrate the effectiveness/ efficacy.

**Questions:**

In section 4, the forward pass of the policy $\pi$ can be better elaborated. For example, with $\pi(b,\omega|s)$ as a map from state to the weight/ bias to generate the cuts, how the input $s$ is input to the GNN-based policy network $\pi$, how the messages pass within the nodes of GNN, and how they eventually lead to the output $\omega, b$?

The approach is implemented to solve optimation problems. It will be helpful to compare it with the vast literature of learning to learn (by RL), e.g., \cite{lan2023learning}.

As the paper is phased as "safe reinforcement learning", it will be helpful to compare it to other safe RL approaches, such as CPO (cited) for decision-making in dynamical systems.

"Provably safe" appears in the title. However, there is no proof included in the paper. Intuitively in this paper's settings, the safety constraints/ model are known, and it is enforced during each iteration, which will lead to safety at each iteration during training.
a) This is a strong prerequisite and makes it unfair to compare with a model-free approach like CPO by claiming that they cannot maintain safety during training.
b) With the models known, how does it compare to many safety projection-based approaches, like those based on control barrier functions \cite{emam2022safe}?
c) With decomposition, will the newly added cuts lead to infeasibility, either in sub- or master problems?



Disclaimer: the reviewer is the author of \textbf{NONE} of the listed literature below. The papers below are just representations of the vast literature in their respective categories.

@article{lan2023learning,
  title={Learning to optimize for reinforcement learning},
  author={Lan, Qingfeng and Mahmood, A Rupam and Yan, Shuicheng and Xu, Zhongwen},
  journal={arXiv preprint arXiv:2302.01470},
  year={2023}
}

@article{emam2022safe,
  title={Safe reinforcement learning using robust control barrier functions},
  author={Emam, Yousef and Notomista, Gennaro and Glotfelter, Paul and Kira, Zsolt and Egerstedt, Magnus},
  journal={IEEE Robotics and Automation Letters},
  year={2022},
  publisher={IEEE}
}

---

### Official Review · Reviewer_Xrr8 · 2024-11-04

**Soundness:** 2
**Presentation:** 3
**Contribution:** 2
**Rating:** 3
**Confidence:** 3

**Summary:**

This paper introduces Bender’s Oracle Optimization (BOO), a method that ensures safety and feasibility in reinforcement learning applications during training and deployment, given access to a feasible set representation (e.g., an imperfect simulator). BOO excels in cases with simple yet inaccurate models, achieving a 17% improvement in job-shop scheduling and near-optimal results in a complex packing problem with 800x faster convergence.

**Strengths:**

1. The paper explores safety in reinforcement learning through the novel Bender's Oracle Optimization, offering a fresh and intriguing approach.
2. The experimental results demonstrate significant performance improvements, highlighting the effectiveness of the proposed method.

**Weaknesses:**

1. Despite the title suggesting "provable" guarantees, the paper lacks concrete theoretical assurances, such as bounds on regret or sample complexity.
2. The experiments compare the proposed method with only a limited selection of prior work, lacking broader validation across diverse environments.

**Questions:**

I noticed the authors mention that their method is a generalized version of constrained reinforcement learning. Could you clarify whether CMDP (Constrained Markov Decision Process) or other classical constrained models in RL are special cases of your model? What is the relationship between them?

**Details Of Ethics Concerns:**

No concerns

---

### Official Review · Reviewer_o69h · 2024-11-08

**Soundness:** 1
**Presentation:** 2
**Contribution:** 3
**Rating:** 5
**Confidence:** 4

**Summary:**

This paper considers Blender’s decomposition oracles to incorporate safety and feasibility constraints for reinforcement learning reward maximization. The authors utilize decomposition to solve subproblem optimization with modified objectives for the master problem instead of relying on constrained MDP. This assumes that the constraints are available during training, deployment, and inference.

**Strengths:**

Utilizing Blender's decomposition oracles for breaking down RL into subproblem and master problem, assuming constraints are available is an excellent idea. This is impactful in safe RL and hierarchical RL.

**Weaknesses:**

Comparison of how the proposed method performs compared to action masking, safety-critics, and other state-of-the-art constrained policy optimization methods is missing and needs further exploration.

The authors mention real-world applications like autonomous driving, but there is no proof of the model's performance on safe RL problems. The approach to incorporating constraints in real-world problems should be elaborated.

The discussion on how the reward and global maximum differ at the beginning of training, and the interpretability of this aspect, needs more detail.

**Questions:**

What would be the performance of BOO in environments with even higher uncertainty? I understand that you mention this as a direction for future work, but could you provide insights on how BOO might handle two distinct levels of stochasticity in practical terms?

After decomposition, is the resulting problem an MDP or a POMDP?

---

### Official Review · Reviewer_nuNB · 2024-11-09

**Soundness:** 3
**Presentation:** 2
**Contribution:** 3
**Rating:** 3
**Confidence:** 3

**Summary:**

This paper presents an RL-based method to solve optimization problems via Bender's decomposition. Specifically, the process of generating Bender's cuts is modeled as a sequential decision-making problem, where the state space includes the current problem solution, variable bounds, and additional auxiliary information; the action space models the weight and bias of a Bender's cut; and the transition function is an existing optimization solver. The authors demonstrate improved performance of this method on a nonconvex constrained problem (where it achieves 2% optimality gap and 850x speedup compared to a global MINLP solver) and a computational job scheduling problem with stochasticity (where it outperforms an MILP solver, with larger relative improvements as stochasticity increases).

**Strengths:**

* This paper cleverly frames Bender's Decomposition as a sequential decision-making problem, and proposes a method to accelerate the Bender's Decomposition solution process using RL (by using RL to efficiently generate Bender's cuts).
* The method achieves significantly improved performance over traditional solvers. Specifically, in the nonconvex constrained problem setting, the method achieves 2% optimality gap and 850x speedup compared to a global MINLP solver. In the computational job scheduling problem with stochasticity, it outperforms an MILP solver (and the fact that it can handle stochasticity is a big strength).

**Weaknesses:**

* My chief complaint is that while the paper is framed in terms of safe RL, the contribution does not strike me as being part of the safe RL literature at all. Instead, it seems fairly squarely positioned within the learning-to-optimize literature, where ML is used to solve optimization problems (that are solved offline, rather than online as sequential decision-making problems). Here, while the _solver_ is framed using sequential decision-making, the original _problem_ being solved is an offline optimization problem. This has several consequences, notably:
    * The related work that is discussed is in the area of safe RL, and no literature on learning-to-optimize is discussed at all. As such, it is hard to fully gauge the contributions of this paper in comparison to prior work.
    * None of the baselines compared against in the experiments are learning-to-optimize baselines.
    * Claims of avoiding catastrophic failures during training and execution seem out of place, since the optimization problems are being solved offline rather than online.
* The method has several assumptions, which are hidden throughout the paper. The implications of these assumptions should be discussed more explicitly:
    * Line 170: Consideration only of optimality constraints (not feasibility constraints)
    * Line 213: Consideration of only linear constraints. While the claim is that this method can handle more general classes of problems, it is worth noting that Bender's Decomposition only applies to certain classes of optimization problems (and while Generalized Bender's Decomposition exists, it is possible the existing method does not work out-of-the-box with that formulation).
* In the experiments, it seems that "default" traditional optimization solvers are considered in the comparisons, but Bender's decomposition _without_ RL does not seem to be considered. Without that comparison, it is difficult to gauge whether the proposed method (as opposed to simply Bender's decomposition itself) is yielding the majority of the reported improvements.
* Minor:
    * In Equation (8), the variables seem to have been flipped compared to Equations (5)--(7).
    * Lines 185 and 186: Typesetting issue with $y^*$
    * Line 196: I believe there is a missing $\mathcal{A} \in$
    * Figure 2: What does the $5.4 \times 10^4$ in the x-axis label indicate?

**Questions:**

* What is the positioning of this work compared to the learning-to-optimize literature (including the part of that literature that uses RL to improve offline optimization solvers)?
* How do learning-to-optimize methods and stronger traditional optimization methods (e.g. pure Bender's decomposition) perform in the experimental settings?
* Over how many seeds are the experimental results run?
* What are the implications of the assumptions made in the work -- specifically, consideration of only optimality cuts (not feasibility cuts) and the use of Bender's decomposition (which addresses only certain classes of optimization problems)?

---

### Official Review · Reviewer_rJLX · 2024-11-09

**Soundness:** 1
**Presentation:** 1
**Contribution:** 1
**Rating:** 3
**Confidence:** 2

**Summary:**

The paper studies how to solve constrained Markov decision processes (MDPs) with safety guarantees during training and deployment. The authors extends the Bender's oracle optimization to constrained MDP problems by assuming a constrained world model. The authors show the effectiveness of this method in a synthetic combinatorial problem and a scheduling problem.

**Strengths:**

- The authors study how to ensure safety constraint satisfaction during training and deployment by assuming a constrained world model. I recongize this is a new constrained learning setting.

- The authors apply the Bender's oracle optimization to constrained MDP problems by enforcing hard constraints during training and inference. This generic framework is very general for handling constraints in RL.

- The authors also provide two computational examples to illustrate potential benefits of the proposed method.

**Weaknesses:**

- The main idea of Bender's oracle optimization is from optimization. The authors directly apply this idea to the constrained MDP problems. This application is incremental to the literature since methods and problems already exist and modifications are not clearly stated.

- I can't find the details on how the constrained MDP problems are solved and what are extra hard constraints. It would be helpful if the authors could present the training algorithm and inference step.

- The convergence rate and sample complexity of  Bender's oracle optimization are not studied. There is no theoretical analysis for understanding how and where Bender's oracle optimization converges. The authors claim provably safe reinforcement learning in the title, which is not evident.

- It would be useful if the authors could provide some scenarios where a constrained world model is available.

- Two experiments are for two classical problems that can be solved by many existing solvers. The authors haven't made a comparison with existing constrained RL methods that are designed for hard constraints.

**Questions:**

see comments in Weaknesses.

---

### Note · Authors · 2024-11-20

**Comment:**

First and foremost, we want to thank all reviewers for reading our paper and the valuable feedback given.

It is clear that we have to work on the presentation and details of our method, therefore we have decided to withdraw our paper for revision and re-submission on a later date.

**Withdrawal Confirmation:**

I have read and agree with the venue's withdrawal policy on behalf of myself and my co-authors.